# Effects of Plant Communities on Human Physiological Recovery and Emotional Reactions: A Comparative Onsite Survey and Photo Elicitation Study

**DOI:** 10.3390/ijerph19020721

**Published:** 2022-01-10

**Authors:** Yifan Duan, Shuhua Li

**Affiliations:** 1College of Landscape Architecture and Arts, Northwest A&F University, Xianyang 712100, China; dyf2010011252@163.com; 2College of Landscape Architecture, Tsinghua University, Beijing 100084, China

**Keywords:** plant community, landscape perception, physical and emotional reaction, physical and mental, environmental restoration effect

## Abstract

We investigated the effects on humans, in terms of skin conductance levels (SCLs) and positive and negative affect schedule (PANAS) scores, of plant communities that differed in their vegetation structure (single-layer woodland, tree-shrub-grass composite woodland, tree-grass composite woodland, and single-layer grassland) through two perceptual methods: onsite surveying and photo elicitation. The results showed that (1) the choice of perception method significantly impacted the PANAS scores of the participants but had no influence on the SCL and (2) viewing a single-layer grassland reduced the SCL (representing the physiological stress level) and improved the positive affect score. The recovery effects for the four vegetation communities were ranked in the order of single-layer grassland > tree-shrub-grass composite woodland > single-layer woodland > tree-grass composite woodland. (3) Gender and professional background significantly impacted the plant community perception methods and landscape experience, and negative affect scores were lower for male participants than for female participants. Participants without backgrounds in landscape design exhibited higher positive affect scores under photo elicitation. Based on the conclusions drawn above, the onsite survey is preferable between the two perception methods. It is recommended that in future landscape designs, combinations of plant community types should be reasonably matched through onsite perception. These research results can provide a scientific basis for the future design of landscapes based on perception experience.

## 1. Introduction

With the development of urbanization, understanding of the public’s views and preferences with regard to urban green space has become indispensable for human-centered design to promote well-being and quality of life [1,2]. Although some findings of landscape perception and experience research have been directly applied in practice, there are doubts about the reliability of the strategies adopted by users, planners, and practitioners, which may lead to a mismatch between the public demand for green space and the actual design of a city [3]. For people living in an urban environment, urban green space is an important element of well-being, but it is often in short supply. One important element for resident well-being and quality of life is the availability of urban green space. There are different ways in which urban green space can positively influence well-being and health [4]. Benefits can accrue from increased activity levels as a result of being in contact with nature [5]. It is important to study the interaction between humans and the objective environment and to integrate this study with the theory of landscape aesthetics to investigate the degree of human satisfaction with and perception and experience of the environment [6]. The study of human perception and experience of landscapes has never ceased [7]. Relevant studies have explained landscape assessment, planning, and design from different perspectives, which have interacted to form a broad field with various theoretical directions and methods [8]. Landscape perception and experience, to a large extent, depend on human visual perception. However, due to the variety of visual strategies used in this study, the failure to test the scientific validity of these strategies will lead to inconsistencies or contradictions in the results and thus to potential risks during application [9]. Therefore, the accuracy and equivalence of various visual strategies for landscape perception and experience should be cross-tested [10,11].

### 1.1. Two Visual Experiences: Onsite Survey and Image Perception

During landscape experience, visual perception is the most important input for perception and evaluation of a landscape by visitors and plays an indispensable role in constructing the visualization framework for the study of green space. Current methods of researching landscape experience usually emphasize the public’s preference for landscapes from the perspective of visual experience through sensory stimulation. Various visual perception methods have emerged to assess the experience of different types of landscapes. Visual methods applied during previous studies can be divided into two categories: image perception and onsite surveys. Using onsite surveys implies that a subject’s perceptual experience is affected by the dual effects of the onsite environment and landscape [12], while image perception signifies that the perceptual experience of a participant is evoked through stimulating media (such as photos) and is usually performed in indoor spaces [13]. If applied separately, these two methods have advantages and disadvantages. Image perception is easy to assess through a presentation of the characteristics of a landscape environment by means of slides and photographs. However, some scholars have seriously questioned the effectiveness of this image perception method in landscape evaluation [14,15]. Others show a positive attitude toward image perception methods and maintain that they can be used as substitutes for onsite surveys [16,17]. T. R. Stewart et al. (1984) argued that photographs can replace onsite surveys in a visual environment assessment [18]. Sevenant and Antrop found the effectiveness of using photographs instead of real landscapes for landscape assessment to be influenced by issues such as perspective and the quality of the photographs taken [3]. However, other researchers hold neutral or negative attitudes toward this method and critique its effectiveness [19,20]. Although onsite and off-site perception methods have been widely used in landscape perception and experience research, questions remain about which approaches are more reliable and whether onsite methods can be successfully replaced. Some results have also shown that participants devoted different degrees of attention to the same scene when viewing the real scene and the photos [21]. Other studies have shown significant differences in their preferences for urban green spaces with different vegetation structures through virtual reality technology, and semi open green spaces receive the highest preference score [14].

The onsite survey approach features the advantages of having strong interaction and a multisensory experience. Combined with questionnaires, interviews, or other techniques (e.g., physiological indicator monitoring), this option can reflect the true feelings of the participants in the environment [22,23]. Onsite surveying is a direct and effective method for assessing landscape experience, allowing the public to obtain a more realistic and effective diversified sensory experience [24,25]. Due to the inconsistency of results among related studies, it is necessary to examine the differences between visual perception methods.

### 1.2. Different Types of Landscape Perception Experience

When people find that they can alleviate stress and fatigue through contact with green space, they begin to be more eager to experience green space, and demand for this space increases [5,26]. Previous studies have attempted to compare the physical and mental recovery effects of green spaces with those of urban environments. A short stay in an urban green space can increase a person’s positive affects (PA) and significantly alleviate negative affects (NA) after stimulation [4,27]. Ulrich et al. (1991) continuously monitored a series of physiological parameters in six different natural and urban environments. They found that participants who watched a stress-inducing movie showed faster and more complete physiological recovery when exposed to natural environments than when exposed to urban environments [28]. In the short term, viewing natural landscapes alleviated the pressure on the participants, and the researchers also found that large urban parks and urban woodland landscapes (area ≥5 hectares) exhibited very similar levels of positive impacts [29]. The skin conductance level (SCL) is a measure of sympathetic nerve activity as reflected by electrical impulses at the skin surface and sweat glands. These activities are controlled only by the sympathetic nervous system (SNS) [30,31,32], and its main function is to stimulate the peripheral nervous system and induce stress responses [33]. Therefore, many studies have used SCL as a measure of physiological health [34]. Studies by Martens (2011) and Gatesleben (2013) have shown that the recovery of mental health is related to various visual perceptions of different natural environments [35,36]. Chiang et al. (2017) reported that the PA of subjects inside a forest was significantly higher than those outside or around the forest, while their NA was greater outside the forest [37]. Similarly, Van den Berg et al. (2014) showed a significant pairwise difference in recovery from NA after exposure to urban streets and parks [1]. For urban green spaces, the combination of plant communities is one of the key factors that affect the perceptual experience of visitors [38,39] which is crucial for designing a diverse urban green space. However, many studies have selected different types of urban green spaces (e.g., woodlands, parks, gardens, wetlands, etc.) to determine the views and preferences of visitors [40,41]. Guiding urban green space planning and design may thus be difficult in practice because the topography of each type of green space is very different [36,42,43]. Therefore, in this study, we consider the combination of plant communities to select the scene.

Due to the systematic field survey, the study of the relationship between the perceived experience of plant community landscapes and health must be further expanded. This investigation is aimed at exploring the differences in landscape experience induced by plant community types through the two perception evaluation methods of onsite surveying and photo elicitation and explores whether there are impacts from the gender or professional background of the participants. SCL measurements and positive and negative affect schedule (PANAS) scores were used to monitor the physical and mental recovery of the participants and were then used to evaluate the experience of perceiving plant community landscapes, thereby providing a scientific basis for future landscape design based on perceptual experience. Our study is aimed at addressing the following questions:(1)How do the two perception methods affect the participants’ physical and mental recovery?(2)How does the type of plant community affect the physiological recovery and emotional reaction of the participants?(3)Are there differences in physical and mental impacts based on the gender or professional background of the participants?

## 2. Materials and Methods

### 2.1. Study Area

The field survey of Beijing urban parks in the early stage showed that the planting forms of single-layer grassland, single-layer woodland, tree-shrub-grass composite woodland, and tree-grass composite woodland were relatively common. The interview conversations with park visitors revealed that these four types of plant communities are highly favored, and the frequency of activities performed by visitors in their surroundings is relatively high. Therefore, these four types of plant community landscapes are typical and representative of urban parks.

Due to the high number of plant species and various types of green spaces, the types of plant species involved in this study were defined as follows. The types of plant communities mentioned in this paper are plant community landscapes common in urban parks and they are composed of three vegetation layers: trees, shrubs, and grass. Plant species include native tree species common in Beijing. Based on the spatial structure of vegetation landscapes and combinations of plant communities, four plant community landscape types were selected: single-layer grassland (*Poa annua* L.), single-layer woodland (*Platycladus orientalis*), tree-grass composite woodland (*Styphnolobium japonicum*, *Populus tomentosa*, and *Poa annua* L.), and tree-shrub-grass composite woodland (*Platycladus orientalis*, *Pinus bungeana*, *Lonicera maackii* (*Rupr*.) *Maxim*, and *Poa annua* L.) (Figure 1).

### 2.2. Experimental Design

#### 2.2.1. Introduction to the Experiments

We designed a randomized experiment to investigate the effects of viewing different types of plant communities in landscaped environments on physical and mental recovery by using the two perception methods. The participants first performed a mathematical test in a noisy environment to induce mental stress, and then each participant was randomly assigned to one of the following eight scenarios: single-layer grassland (onsite/photo), single-layer woodland (onsite/photo), tree-shrub-grass composite woodland (onsite/photo), or tree-grass composite woodland (onsite/photo), as shown in Figure 2. The participants were immersed in this environment for 5 min. To evaluate and compare their physiological stress responses, we continuously measured the SCLs in these plant communities during the experiment. We also used the PANAS assessment to compare their positive and negative affective states [44] and evaluated them at three different time points during the experiment (Figure 3).

#### 2.2.2. Procedure

First, we briefly explained the experiment to the participants and obtained written informed consent. Then, the participants filled out a short demographic questionnaire and were taken to the experimental scene area. In the scenic area, we connected an electrode to each participant’s skin to measure their SCL. The SCL was continuously monitored and measured throughout the rest of the experiment. Next, the participants were asked to calm down for 5 min. Their average SCL during this period was used as the SCL baseline. They then completed the preliminary PANAS assessment, and the assessment result was the benchmark for assessing the impacts; this stage was denoted as T0. They were then asked to complete a 5-min mathematical operation test to stimulate stress [44], and their average SCL during the test was used to represent their SCL under stress. The participants then completed the second PANAS assessment, which was denoted as T1. Each participant was then randomly assigned to one of the four scenes. In this environment, each participant was asked to wear homemade viewfinder glasses to limit the field of view. Participants assigned to photo elicitation were asked to wear noise-reducing earphones to reduce the influence and interference of external environmental factors. The participants were required to watch the scene for 5 min [13], and the average SCL was measured and recorded. After viewing the scene, the participant completed the third and final PANAS assessment, which was denoted as T2, after which the SCL measurement equipment was removed, and the participants left the test site.

(a)Onsite surveys

Onsite surveys were performed in clear weather and at moderate temperatures (no rain). The average temperature of the sites was 19.7 °C (19.65 ± 2.22), and the relative humidity averaged 53.4% (53.35 ± 8.26). In addition, before the start of each test, a reminder was posted within 2 m of the test site to inform visitors of the test area, thereby reducing interference from external factors (such as visitors’ activities, noise, etc.). To reduce the occurrence of confounding variables, we ensured that the surrounding environment was quiet while keeping the light, temperature, humidity, and wind speed in the landscape area similar (Figure 3a).

(b)Photo elicitation

The image perception test was performed using a digital camera (Thailand SONY ILCE-5000) to take two-dimensional color photos, which were captured under sunny weather and moderate temperatures (no rain) (Figure 3b). To ensure that all the landscapes were included in the scene, four photos were taken at each scene, and the shooting angle was 45°/frame. When taking photos, the aperture, shutter, and ISO were set to F4, 1/160 s, and 200, respectively, and character interference was prevented by not taking pictures with people in the frame (Figure 4). Photo elicitation was conducted in quiet and well-lit classrooms.

#### 2.2.3. Vegetation Period

The selected landscape study area is within the vegetation period (autumn). The vegetation period in the onsite surveys was approximately 10 October 2019, and the test time was between 10 October and 20 October. The photos taken for the remote surveys were from 11 October to 18 October 2019. The photo elicitation experiment was completed in quiet and well-lit classrooms from 2 November 2019 to 15 November 2019.

### 2.3. Participants

A total of 320 college students (M age = 22.03, SD age = 2.13, age-range = 18–25 years) studying various subjects were recruited as the participants in this study and were divided into a professional group that studied the subject of landscape architecture, and a nonprofessional group that included students who studied other subjects (curriculum disciplines not including mathematical calculation). The ratio of males to females was 1:1, and the ratio of professionals to nonprofessionals was 1:1.5. All the participants were healthy students and were Chinese speakers. All the participants were informed of the trial procedure, related risks, and confidentiality issues, and all participants signed an informed consent form before starting the experiment. This study was conducted according to the Declaration of Helsinki. Participants were randomly divided into eight groups, with 40 individuals in each group. We randomly assigned each participant to one of the eight groups corresponding to combinations of plant community type and measurement technique (Figure 3). After the test was completed, each participant received a payment for their participation.

### 2.4. Stress Task

To stimulate stress, the participants were asked to complete a 5-min numerical calculation test. We told the participants that the purpose of this experiment was to evaluate their performance in numerical calculations, and we scored and ranked their performance to reflect the participants’ physiological recovery and emotional changes more clearly after the experience. We used numerical calculations and simulations of noisy environments to stimulate psychological and physiological responses. Previous studies have demonstrated the effectiveness of stressors in a noisy environment either by SCL or negative affects (NA), indicating that stressors can increase SCL or depress mood, respectively [45,46].

### 2.5. Measurement

#### 2.5.1. Physiological Measures

We used a biofeedback device in a Biopac MP 150 (*MP Systems with AcqKnowledge, USA*) workstation and its accompanying AcqKnowledge (*MP Systems with AcqKnowledge, USA*) software to monitor, measure, and record the SCL values. Skin conductance is usually employed as a measure of pressure. A significant increase in SCL indicates a pressure change from a stable state to a state of stress and tension (which is usually related to an increase in sweat), and a decrease in SCL indicates stress relief and a gradual shift to a relaxed and calm state [29].

#### 2.5.2. Psychological Measures

The subjective influence was evaluated using the PANAS [44], which included 10 words representing positive words and 10 words representing negative words. Positive vocabulary scores add up to PA scores, negative vocabulary scores add up to NA scores, and PA and NA scores respond to positive and negative changes in an individual’s affectation, respectively [45]. We performed the PANAS assessment before and after the mathematics test and after the viewing immersion. This instrument showed high reliability and the Cronbach’s α score ranged between 0.84 and 0.88.

### 2.6. Data Analysis

The data obtained here were processed using IBM SPSS Statistics 26.0. (IBM, Inc., Armonk, NY, USA) The timeline was divided into three segments. T0 was the baseline, T1 was the stress stage, and T2 was the immersion stage. To test the differences in the environmental recovery effect, we calculated the changes in the parameters with the following formulas:Stress value ΔT1 = T1 − T0,
Immersion value ΔT2 = T2 − T0,
Recovery value ΔT3 = ΔT2 − ΔT1.

To test whether the two perception methods and the four plant community types affected physical and mental recovery, we used a one-way analysis of variance (ANOVA) to compare the recovery values (ΔT3) of the different types of plant communities for the two perception methods. To determine whether each type of plant community landscape influenced psychological change, we used paired sample t-tests to compare PA and NA scores before and after the experience. Finally, we used one-way ANOVA to test the effects of gender and professional background on the physical and mental recovery of participants experiencing the plant community types using the perception methods.

## 3. Results

### 3.1. Impacts of Perception Methods and Plant Community Types

#### 3.1.1. Impacts of Perception Methods

According to the results of the one-way ANOVA, there was a significant difference in recovery between the onsite survey and photo elicitation methods, but we only found a significant difference in the NA (Table 1). The NA recovery value for the onsite surveys was smaller than the NA recovery value for photo elicitation. This finding indicates that the onsite survey perception method alleviates the NA of the participants.

#### 3.1.2. Impacts of Plant Community Types

The analysis of the recovery values (ΔT3) for various indicators of the landscape experience in the four types of plant communities showed that the type of plant community significantly influenced the recovery of emotional reactions but had no effect on the SCL. There were only four significant differences in PA: between single-layer grassland and single-layer woodland (*p* = 0.006), between single-layer grassland and tree-grass composite woodland (*p* = 0.008), and between single-layer woodland and tree-shrub-grass composite woodland (*p* = 0.043). Among them, the single-layer grassland exhibited the greatest recovery value, followed by the tree-shrub-grass composite woodland, the single-layer woodland, and the tree-grass composite woodland.

There were only significant differences in the NA for three comparisons: between single-layer grassland and tree-shrub-grass composite woodland (*p* = 0.047), between single-layer grassland and tree-grass composite woodland (*p* = 0.002), and between single-layer woodland and the tree-grass composite woodland (*p* = 0.019). Among these, the single-layer grassland featured the largest recovery value, followed by the single-layer woodland, the tree-shrub-grass composite woodland, and finally the tree-grass composite woodland (Table 2).

### 3.2. Differences in the Physiological Recovery of Participants Experiencing Different Plant Community Types

#### 3.2.1. Onsite Surveys

An analysis of the chart of SCL trends (Figure 5 and Table 3) shows that the onsite surveys in single-layer grassland elicited increases throughout the first 10 min, and the SCLs of the participants began to increase gradually during the stress stage (ΔT1) and began to decline rapidly at the 10th minute, followed by stability for 5 min. The stress values of participants who experienced the single-layer woodland also continued to increase. A relatively obvious decrease occurred during the following 5 min, but during the last 3 min, the stress values began to increase. The SCLs of participants fluctuated during the stress stage (ΔT1) and the immersion stage (ΔT2) when experiencing the tree-shrub-grass composite woodland. Following 2 min of viewing, the SCL values of the participants continued to decrease. The SCL values decreased most significantly after the onsite surveys in the single-layer grassland, followed by those of participants experiencing the tree-shrub-grass composite woodland and the single-layer woodland, with the smallest reductions in SCLs observed for participants experiencing the tree-grass composite woodland.

#### 3.2.2. Photo Elicitation

Among the plant community landscapes perceived through photo elicitation, the SCL trend in the tree-shrub-grass composite woodland was more obvious, and the SCLs of the four plant community landscapes maintained an upward trend during the stress stage and continued to increase during the stress stage for all four plant community landscapes. During the immersion stage, the SCLs of participants viewing the four plant communities all declined, but the SCL values of the participants remained stable when they were experiencing the tree-shrub-grass composite woodland (Figure 6 and Table 4). The SCL values declined most significantly after the participants viewed the tree-shrub-grass composite woodland, followed by those of participants viewing the single-layer grassland and single-layer woodland, and those of participants viewing the tree-grass composite woodland decreased the least.

### 3.3. Differences in the Emotional Reaction Recovery of Participants Experiencing Different Types of Plant Communities

#### 3.3.1. Onsite Surveys

Based on the affect scores of the participants after experiencing the four plant community landscapes, the PA scores were found to be higher after the participants experienced the single-layer grassland during the immersion stage (ΔT2) than scores measured during the stress stage (ΔT1); however, the PA scores of participants experiencing the other plant community landscapes decreased (Figure 7 and Table 5), but there were no significant differences between the stages. This finding indicates that after viewing the single-layer grassland, the participants’ PA values were enhanced.

Compared with the NA scores measured during the stress stage (ΔT1), the NA scores of all the participants decreased after they viewed the plant community landscapes during the immersion stage (Δ2), indicating that the NA of participants was alleviated after viewing the plant community landscapes. Among these results, the decline in NA was greatest after viewing single-layer grassland, followed by single-layer woodland, tree-shrub-grass composite woodland, and tree-grass composite woodland, and the reduction was smallest for the tree-grass composite woodland (Figure 8 and Table 6).

#### 3.3.2. Photo Elicitation

After the participants experienced the plant community landscapes by photo elicitation, patterns of change in the PA scores from the stress stage (ΔT1) to the immersion stage (ΔT2) differed depending on the plant community in view (Figure 9 and Table 7). After the viewing of the grassland and tree-shrub-grass composite woodlands, the PA scores of the participants increased.

Compared with the scores measured during the stress stage (ΔT1), the NA scores of all the participants declined after they viewed the plant community landscapes during the immersion stage (ΔT2), indicating that after viewing the plant communities, the NA of participants was alleviated. Among these observations, the NA of participants decreased significantly after they viewed the single-layer grassland, followed by lesser effects for the single-layer woodland and the tree-grass composite woodland, with the smallest reduction observed for the tree-shrub-grass composite woodland (Figure 10 and Table 8).

### 3.4. Effects of Perception Methods and Plant Community Types on the Physiology and Psychology of Participants of Different Genders and Professional Backgrounds

#### 3.4.1. Impacts on Participants with Different Professional Backgrounds of Perception Methods

The analysis showed that there was a significant difference in the PA between participants with different professional backgrounds (*p* = 0.004), but there was no effect of gender (Table 9). The PA scores of the participants were higher for the professional group than for the nonprofessional group. We also found that the PA (*p* = 0.003) and NA (*p* = 0.009) of participants with different professional backgrounds were significantly different between the two perception methods. The onsite survey and photo elicitation methods showed the opposite trends in the PA and NA of the participants with different professional backgrounds. Participants in the professional group showed higher PA scores during onsite surveys, while those in the nonprofessional group displayed higher PA scores under photo elicitation. The two patterns for NA were just the opposite. Participants in the nonprofessional group had lower NA scores under the onsite survey, while those in the professional group exhibited lower NA scores under photo elicitation (Table 2).

#### 3.4.2. Effects of Plant Community Types on Participants of Different Genders

The analysis showed that the NA differed significantly between participants of different genders. Male participants had lower NA scores than did female participants (Table 10). Viewing plant community landscapes had a greater effect on relieving NA for male participants than female participants (*p* = 0.048).

#### 3.4.3. Effects on Participants with Different Professional Backgrounds of Plant Community Types

Participants with different professional backgrounds showed significant differences in PAs after viewing the plant community landscapes (*p* = 0.014). Participants in the professional group showed higher PA scores after viewing plant community landscapes than did participants with no background in landscape design. Participants in the professional group exhibited higher PA scores after viewing single-layer grasslands, tree-shrub-grass composite woodlands, and single-layer woodlands, while participants in the nonprofessional group displayed higher PA scores after viewing tree-grass composite woodlands (Table 11).

## 4. Discussion

### 4.1. Effects of Two Perception Methods on Participants’ Physical and Mental Recovery

There was a significant difference in NA values between the two perception methods and the mitigation effect of onsite surveying on NA was greater than that of photo elicitation. This result indicates that the onsite survey technique exerts a more obvious effect on alleviating the NA. Therefore, it is necessary to consider the experience of landscape perception in the onsite environment at the beginning of the landscape design process.

There are three possible reasons for the difference in effectiveness between the two methods. First, a potential problem in the integrity of the landscape and the coherence of the picture may be present. Previous studies have shown that photographic landscape images cannot provide complete visual information and imagery provided by the environment [3,46]. In terms of landscape perception and experience, there were still significant differences between photo elicitation and onsite surveying. During onsite surveying of green spaces, participants experience a variety of sensory perceptions through the properties of sound, air humidity, temperature, wind, and light. Additionally, different types of plant communities constitute different spatial structures and have different effects on visitors’ perceptions and experiences [47].

Another possible reason is that during site investigation when a participant is brought to the actual environment of a scene, they can immediately establish an effective connection with the landscape that can be explained by the biophilia hypothesis [48], which states that there is an intrinsic emotional connection between human beings and nature. In terms of the experimental environment, compared to onsite surveys, the indoor experimental environment perceived via photo elicitation was quieter and more private, the SCL and NA in the stress stage continued to increase, and the PA continued to decrease. Thus, the plant community landscape viewing during the immersive stage led to significant effects on the physical and mental recovery of the participants. In terms of the overall effect of the experiment, although the perceptual experience via photo elicitation affects the physical and mental recovery of the participants, a certain gap with the onsite surveying remains.

The last possible reason is that the difference between the dynamic perception and the static perception process leads to a difference in the landscape experience. Kroh and Gimblett (1992) found that dynamic factors may affect multisensory synergistic perception in a landscape [15], thereby modulating the effect of the landscape experience [49]. In our study, participants engaged in onsite surveying were allowed to perform small-scale activities, and they could perceive the environment through multiple senses such as hearing, vision, and smell. However, the participants under photo elicitation were not allowed to walk around, and they could only view the images through their visual sense organs. Therefore, onsite surveying makes the participant’s perception experience better than the image perception experience from visual sensory stimulation alone [50].

### 4.2. Interaction between Physiological and Emotional Reaction Recovery of Participants Experiencing Plant Communities

#### 4.2.1. Effect of Plant Community Type on Participants’ Bodies and Minds

The visual perception of the four plant community landscapes showed no significant impact on the physiological recovery of the participants but significantly improved their emotional reaction recovery. Among the plant communities, viewing a single-layer grassland exhibited the greatest impacts on the participant PA and NA. This plant community was followed by single-layer woodland, tree-shrub-grass composite woodland, and tree-grass composite woodland. The differences in effectiveness among plant community landscapes may be a result of different planting methods and planting layouts forming different enclosed spaces, which bring different perceptual and experiential effects to viewers [51].

The single-layer grassland is an example of open space, and the vision of the participants is open without any obstruction [52,53]. Some researchers have proposed that the reason we are attracted to this environment is that it is a place where humanity has evolved over tens of thousands of years, and “feels at home”. Even in other environments, such as mountains and forests, humans will look for grasslands [54]. The single-layer woodland is an overstory landscape composed of coniferous plants. The prospect-refuge theory proposes that humans prefer two attributes of landscape environments: prospects and refuges, providing lookouts and shelter, respectively, and this type of landscape environment is worthy of attention [55]. The single-layer woodland conforms to the scenario described in prospect-refuge theory. In addition, the volatiles of coniferous plants can relax people’s emotions and help the body and mind recover [56]. The tree-grass composite woodland zone provides visitors with a sense of distance [57]. In tree-grass composite woodland, visitors only feel the prospect, providing a lookout, but the shelter providing protection is missing, and tree-grass composite woodland lacks space for understory activity, which does not meet the psychological needs of humans for shelter, thus affecting the participant experience. The tree-shrub-grass composite woodland provides a space for people to live alone and communicate with each other and ensure their safety.

#### 4.2.2. Interaction between Physiological and Mental Recovery

When encountering a stressful task, the participants initially showed avoidance or dislike [58], which stimulated their SNS stress response and thus increased their SCL. Skin conductance techniques can measure electrical impulses on the skin surface and sweat glands, and these electrical impulses are only controlled and generated by the SNS. We believe that the SCL can reflect the changes in the SNS well. We also found that the changes in stress values from photo elicitation were significantly greater than those from onsite surveys. This finding also indicates that the environment of the experiment hinders the stress responses of the participants. By contrast, the observed responses to stress in the field were less than those indoors, preceding photo elicitation. We also demonstrated the effect of the natural landscape on stress mitigation from another perspective.

The SCLs of the participants decreased in all eight combinations of landscape scenes and perception methods. A possible reason is that avoiding the source of stress is equivalent to avoiding tension and threat. Compared with the stress source, a plant community landscape is comfortable and quiet and therefore can elicit PA responses [58]. During the immersion stage, the PA of the participants reduced the SNS and gradually restored the body and mind, which in turn reduced the SCLs of the participants [58]. The plant community landscapes under these two perception methods, therefore, provided the participants with the opportunity to restore their emotional state, which is consistent with the results of Ulrich et al. (1991) and Alvarsson et al. (2010) [28,31].

### 4.3. Impacts of Perception Method, Plant Community Type, and Gender and Professional Background of Participants on the Landscape Experience

In this study, the gender and professional background of the participants modulated the effects of plant communities on the physical and mental indicators of landscape perception experience. The NA of the male participants was smaller than that of the female participants. Participants lacking a background in landscape design showed a higher PA under photo elicitation, which is consistent with the findings of Wang et al. [59], but is contrary to the findings of other studies [60,61]. Men prefer woodland landscapes more than women do, possibly because women are often afraid of passing through woodland landscapes because women feel fear in remote places [59,62,63]. With regard to professional background, each academic major corresponds to some specific “knowledge”, and this “knowledge” may act as an intermediary variable in the process of preference formation [64], indicating that school education in different majors may be a mechanism for transmitting preferences [65]. The result may be that visitors who rarely see pleasing vegetation landscapes tend to score higher [66]. Moreover, although the selected locations in this study differ in the combination of plant communities, they are all types of plant communities that are common in urban environments and are familiar to students [67,68]. Therefore, to determine the impact of gender and professional background on landscape perception and experience, performing an appropriate analysis is very important in further research.

### 4.4. Future Directions

Our study demonstrates the difference in the physical and mental recovery effects between onsite surveys and photo elicitation and shows that the methods of perceiving each plant community type are complementary to each other and cannot be replaced by one another. In people’s perception, the recovery effect from onsite surveys is far greater than that from the perception of photos. Our results support the argument by de Kort et al. (2006) that virtual natural environments exert a restoration effect similar to that of the real environment [34]. Although the recovery effect from a virtual natural environment is limited, it affects physical and mental recovery to some extent. Therefore, in future landscape design, we may use these two perception methods in combination. Before presenting an actual landscape environment, we could first use the perception method of photo elicitation to allow a participant to experience the landscape and then bring the participant into the real landscape environment. This approach may maximize the landscape perception experience.

Our findings indicate that viewing single-layer grassland leads to an increase in PA, which is also consistent with the theory of Ulrich (1983) [69]. This increase may be a result of participants thinking that the landscape environment of a single-layer grassland can generate more PA because grassland may provide opportunities for walking, playing, or lying down. Therefore, a positive evaluation of this environment will produce the corresponding post-cognitive effect. It also causes post-cognitive arousal (physiological recovery), and in turn, a higher PA reduces the NA brought by coping with anxiety [70]. In future landscape design, the combination of plant community types can be flexibly matched according to the actual environment and scene. For example, a single-layer grassland provides visitors with a space for activities and leisure, a single-layer woodland provides an activity space for exercise, a tree-grass composite woodland provides a recreational space for people to shelter during the summer, and a tree-shrub-grass composite woodland provides a private space for meditation and communication. In different scenes or spaces, the combination and collocation are determined according to the needs of the designer or the user to ultimately create a landscape environment suitable for human activities.

## 5. Conclusions

This study is the first to explore the landscape experience of plant community types using both onsite and image (photo) perception methods. The aim is to provide scientific support for the evaluation of landscape perception and experience effects in the future. There are some notable conclusions. First, the choice of perception method significantly impacted the PA and NA scores of the participants but showed no effect on SCLs. Second, viewing a single-layer grassland reduced SCLs (representing the physiological stress level) and improved PA scores. The single-layer grassland exhibited the strongest recovery effect, followed by the tree-shrub-grass composite woodland; the single-layer woodland showed the least recovery. Participants watched the lawn community in the form of an onsite survey, which reduced SCL and simultaneously had a better inhibitory effect on negative emotions. Furthermore, gender and professional background significantly impacted the effects of perception methods and plant community landscape experience, and the PA scores of male participants were higher than those of female participants. Participants lacking a background in landscape design displayed higher PA under photo elicitation. Finally, based on the conclusions drawn above, onsite surveying is preferable between the two perception methods. Thus, the approach to landscape perception should be carefully selected for a specific landscape during a specific season to provide a scientific basis for evaluating landscape perception and preference in the future.

However, this study has some weaknesses. First, all the participants were college students. Many other investigations have also used college students as the target population, though the results may not fully reflect other social groups. In addition, the age range of participants was between 19 and 25 years old, while other sample age ranges may be different, and the age variable could act as an effect modifying the variable. In future studies, the age variable must be further studied. Therefore, the sample population should be expanded in subsequent studies. Second, our study used the model of stopping to admire a plant landscape, in which the behavioral patterns of participants’ appreciation of the scenery are mostly stopping and touring, which is more consistent with the ways in which visitors experience landscapes when recreating in parks. Therefore, in follow-up studies, we can consider the use of different forms of landscape perception and experience. Finally, this study was only conducted in the autumn, but the preferences of the participants may be affected by the changes in weather and seasonal changes in plants, especially through the perception method of onsite surveying. The landscapes of plant communities during different seasons should be further studied and explored. In addition, the method of showing photographs may be somewhat “single” compared to perceiving the landscape in situ. Perhaps employing a third method, such as the use of virtual reality goggles, could thus increase the richness of the experimental design.

## Figures and Tables

**Figure 1 ijerph-19-00721-f001:**
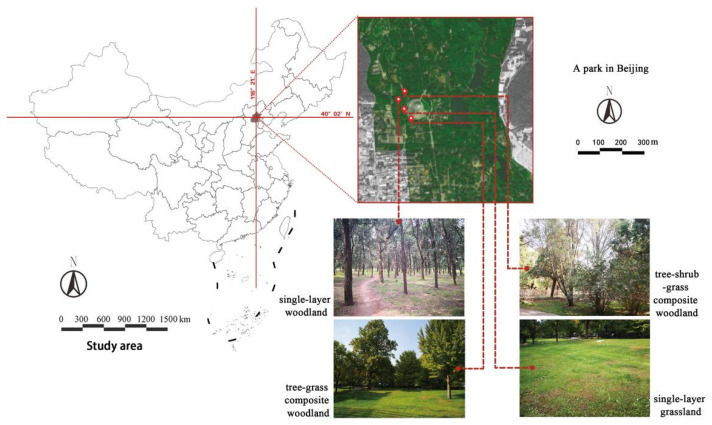
Study area.

**Figure 2 ijerph-19-00721-f002:**
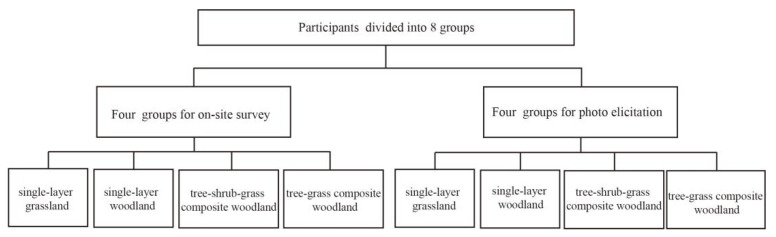
The hierarchical arrangement of participation groups.

**Figure 3 ijerph-19-00721-f003:**
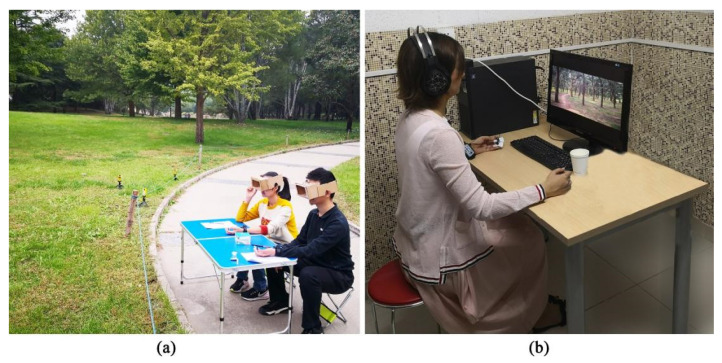
The participants used two methods to perceive different plant community landscapes, (**a**) onsite survey and (**b**) photo elicitation.

**Figure 4 ijerph-19-00721-f004:**
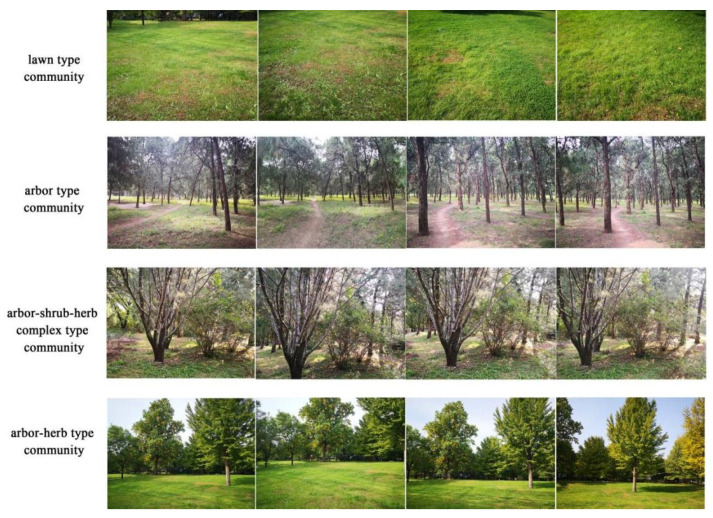
Two-dimensional color photographs of the selected study area.

**Figure 5 ijerph-19-00721-f005:**
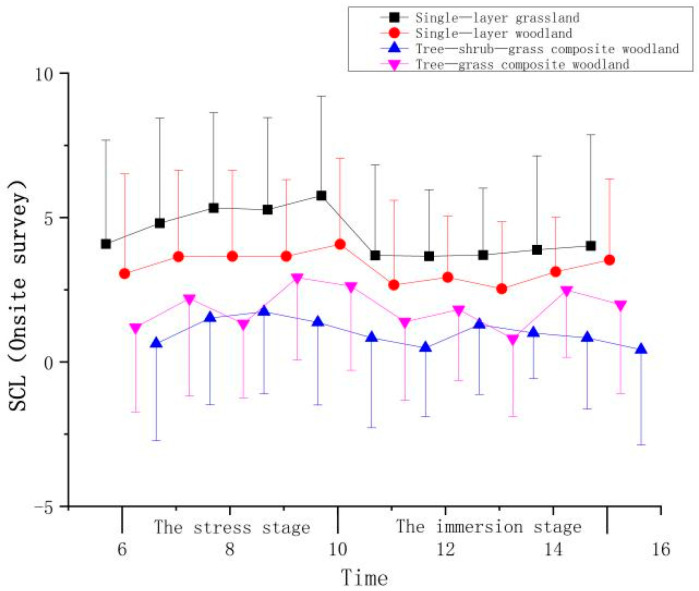
Impact of onsite surveys on participant SCL indicators.

**Figure 6 ijerph-19-00721-f006:**
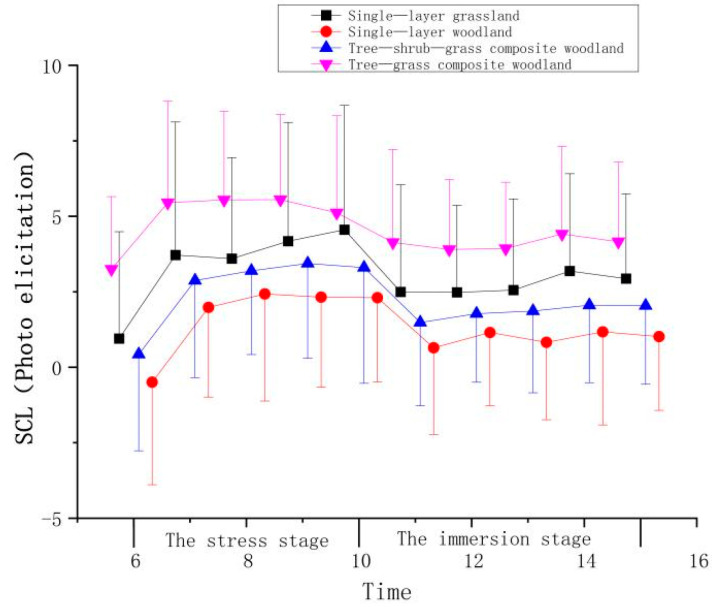
Effect of photo elicitation on the SCL indicators of the participants.

**Figure 7 ijerph-19-00721-f007:**
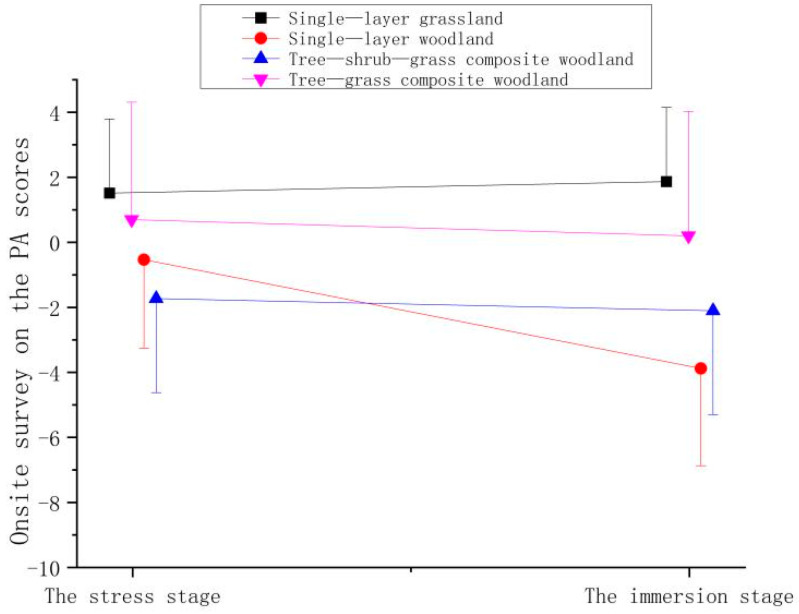
Effect of onsite surveys on participant PA scores.

**Figure 8 ijerph-19-00721-f008:**
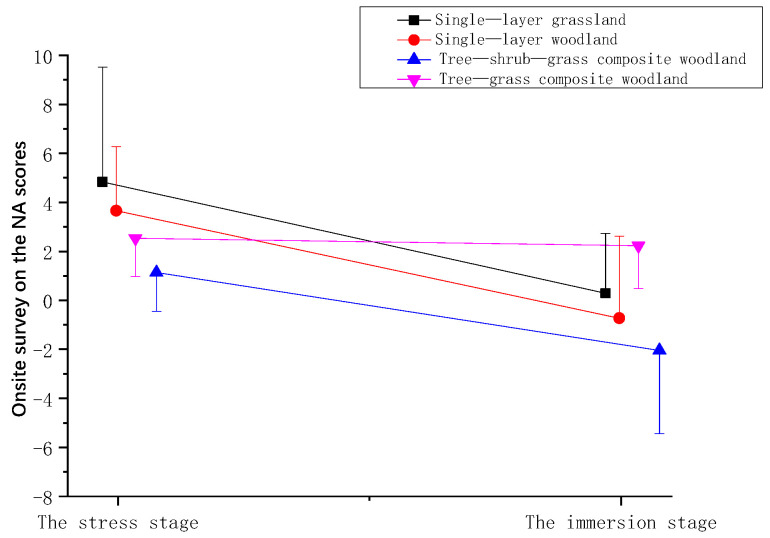
Effect of onsite surveys on the NA scores of participants.

**Figure 9 ijerph-19-00721-f009:**
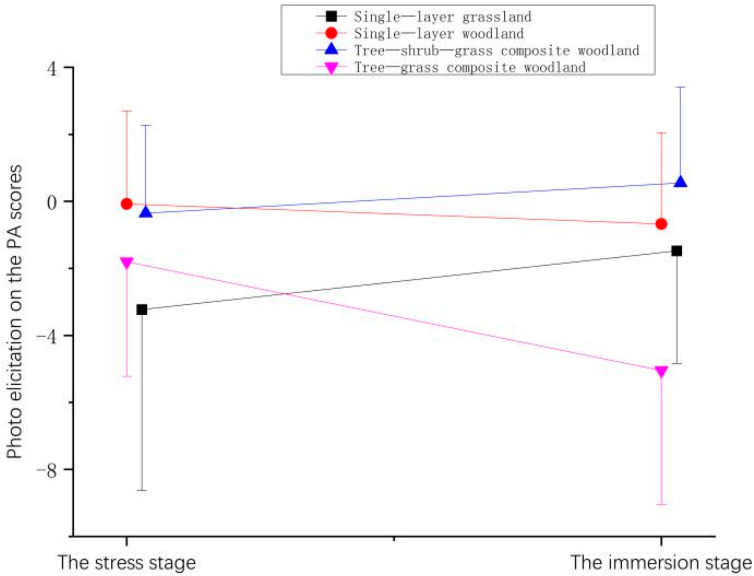
Effect of photo elicitation on the PA scores of participants.

**Figure 10 ijerph-19-00721-f010:**
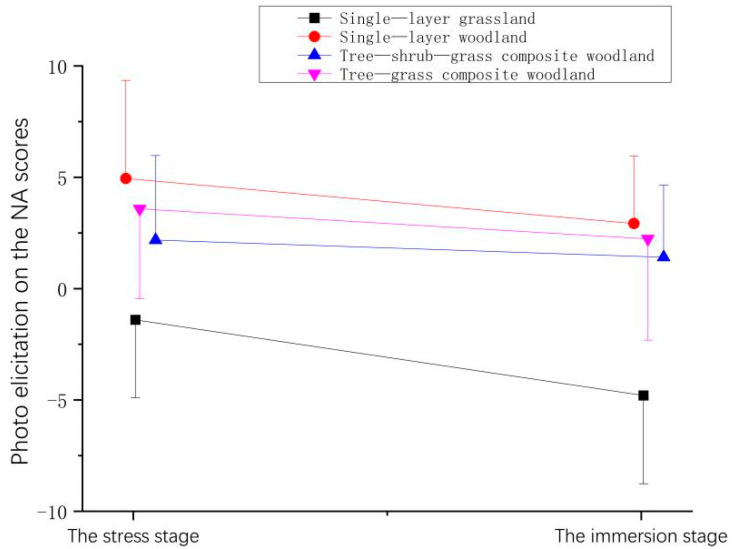
Effect of photo elicitation on the NA scores of the participants.

**Table 1 ijerph-19-00721-t001:** Differences in experience between the two perception methods.

Indicator	Onsite Survey (ΔT3)	Photo Elicitation (ΔT3)	F Value	Significance
SCL	−0.67 ± 0.17	−0.77 ± 0.123	0.129	0.720
PA	−0.97 ± 0.596	−0.3 ± 0.516	0.741	0.390
NA	−3.1 ± 0.507	−1.89 ± 0.507	6.415	0.012

**Table 2 ijerph-19-00721-t002:** Differences in recovery values among different types of plant communities.

Indicator	Single-Layer Grassland (ΔT3)	Single-Layer Woodland (ΔT3)	Tree-Shrub-Grass Composite Woodland (ΔT3)	Tree-Grass Composite Woodland (ΔT3)	F Value	Significance
SCL	−0.96 ± 0.23	−0.71 ± 0.2	−0.61 ± 0.2	−0.62 ± 0.21	0.321	0.81
PA	1.05 ± 0.65	−1.98 ± 1	0.26 ± 0.69	−1.88 ± 0.73	4.924	0.002
NA	−3.98 ± 0.77	−3.2 ± 0.61	−1.98 ± 0.75	−0.83 ± 0.704	5.225	0.002

**Table 3 ijerph-19-00721-t003:** Impact of onsite surveys on participant SCL indicators.

Stage	Time	Mean ± (SD)			
		Single-Layer Grassland	Single-Layer Woodland	Tree–Shrub–Grass Composite Woodland	Tree–Grass Composite Woodland
The stress stage	6	1.83 ± 3.59	1.29 ± 3.45	0.57 ± 3.36	0.73 ± 2.93
	7	2.55 ± 3.63	1.88 ± 3.00	1.46 ± 3.06	1.73 ± 3.39
	8	3.08 ± 3.30	1.89 ± 2.97	1.68 ± 2.85	0.86 ± 2.57
	9	3.02 ± 3.18	1.89 ± 2.65	1.31 ± 2.86	2.45 ± 2.86
	10	3.51 ± 3.44	2.30 ± 2.97	0.77 ± 3.11	2.16 ± 2.92
The immersion stage	11	1.44 ± 3.13	0.90 ± 2.93	0.42 ± 2.39	0.92 ± 2.73
	12	1.41 ± 2.29	1.16 ± 2.12	1.23 ± 2.42	1.35 ± 2.46
	13	1.45 ± 2.31	0.77 ± 2.32	0.94 ± 1.58	0.33 ± 2.69
	14	1.63 ± 3.25	1.36 ± 1.88	0.77 ± 2.47	2.02 ± 2.34
	15	1.77 ± 3.84	1.76 ± 2.81	0.36 ± 3.30	1.52 ± 3.08

**Table 4 ijerph-19-00721-t004:** Effect of photo elicitation on the SCL indicators of the participants.

Stage	Time	Mean ± (SD)			
		Single-Layer Grassland	Single-Layer Woodland	Tree–Shrub–Grass Composite Woodland	Tree–Grass Composite Woodland
The stress stage	6	0.20 ± 3.54	−0.11 ± 3.40	0.01 ± 3.19	0.48 ± 2.40
	7	2.97 ± 4.40	2.37 ± 2.97	2.46 ± 3.22	2.68 ± 3.36
	8	2.85 ± 3.35	2.81 ± 3.55	2.78 ± 2.78	2.78 ± 2.93
	9	3.43 ± 3.93	2.71 ± 2.99	3.03 ± 3.14	2.78 ± 2.83
	10	3.81 ± 4.12	2.69 ± 2.80	2.89 ± 3.84	2.35 ± 3.21
The immersion stage	11	1.75 ± 3.55	1.03 ± 2.87	1.07 ± 2.77	1.37 ± 3.07
	12	1.73 ± 2.88	1.54 ± 2.42	1.36 ± 2.28	1.13 ± 2.31
	13	1.81 ± 3.02	1.21 ± 2.57	1.45 ± 2.72	1.17 ± 2.19
	14	2.44 ± 3.22	1.56 ± 3.08	1.65 ± 2.57	1.65 ± 2.90
	15	2.19 ± 2.81	1.40 ± 2.44	1.63 ± 2.61	1.39 ± 2.64

**Table 5 ijerph-19-00721-t005:** Effect of onsite surveys on participant PA scores.

Stage	Mean ± (SD)			
	Single-Layer Grassland	Single-Layer Woodland	Tree–Shrub–Grass Composite Woodland	Tree–Grass Composite Woodland
The stress stage	−0.18 ± 2.27	−0.50 ± 2.72	−0.80 ± 2.90	−1.18 ± 3.61
The immersion stage	0.18 ± 2.28	−3.85 ± 2.99	−1.18 ± 3.20	−1.68 ± 3.83

**Table 6 ijerph-19-00721-t006:** Effect of onsite surveys on the NA scores of participants.

Stage	Mean ± (SD)			
	Single-Layer Grassland	Single-Layer Woodland	Tree–Shrub–Grass Composite Woodland	Tree–Grass Composite Woodland
The stress stage	3.13 ± 4.68	1.83 ± 2.62	0.28 ± 1.58	0.88 ± 1.56
The immersion stage	−1.4 ± 2.45	−2.55 ± 3.34	−2.90 ± 3.40	0.58 ± 1.75

**Table 7 ijerph-19-00721-t007:** Effect of photo elicitation on the PA scores of participants.

Stage	Mean ± (SD)			
	Single-Layer Grassland	Single-Layer Woodland	Tree–Shrub–Grass Composite Woodland	Tree–Grass Composite Woodland
The stress stage	−3.25 ± 5.41	−0.08 ± 2.77	−0.38 ± 2.62	−1.80 ± 3.42
The immersion stage	−1.50 ± 3.36	−0.68 ± 2.72	0.53 ± 2.85	−5.05 ± 3.99

**Table 8 ijerph-19-00721-t008:** Effect of photo elicitation on the NA scores of the participants.

Stage	Mean ± (SD)			
	Single-Layer Grassland	Single-Layer Woodland	Tree–Shrub–Grass Composite Woodland	Tree–Grass Composite Woodland
The stress stage	−1.35 ± 3.50	2.65 ± 4.40	1.65 ± 3.79	1.58 ± 4.04
The immersion stage	−4.75 ± 3.97	0.63 ± 3.03	0.88 ± 3.24	0.23 ± 4.55

**Table 9 ijerph-19-00721-t009:** Effects of professional background and perception method.

Indicator	Onsite Survey		Photo Elicitation		F Value	Significance
	Professional Group	Nonprofessional Group	Professional Group	Nonprofessional Group		
SCL	−0.81	−0.47	−0.71	−0.87	1.477	0.225
PA	0.97	−3.88	−0.35	−0.22	8.978	0.003
NA	−1.75	−5.13	−2.08	−1.59	6.884	0.009

**Table 10 ijerph-19-00721-t010:** Effects of gender and plant community type on physiological and psychological indicators.

Indicator	Single-Layer Grass		Single-Layer Woodland		Tree-Shrub-Grass Composition Woodland		Tree-Grass Composition Woodland		F Value	Significance
	Male	Female	Male	Female	Male	Female	Male	Female		
SCL	−1.19	−0.74	−0.93	−0.48	−0.94	−0.27	−0.75	−0.48	0.25	0.858
PA	1.3	0.8	−3.58	−0.38	−0.98	1.5	−1.55	−2.2	0.99	0.398
NA	−4.23	−3.73	−3.6	−2.8	−4.28	0.33	−0.98	−0.68	2.669	0.048

**Table 11 ijerph-19-00721-t011:** Effects of professional background and plant community type on physiological and psychological indicators.

Indicator	Single-Layer Grass		Single-Layer Woodland		Tree-Shrub-Grass Composite Woodland		Tree-Grass Composite Woodland		F Value	Significance
	Professional Background	Nonprofessional Background	Professional Background	Nonprofessional Background	Professional Background	Nonprofessional Background	Professional Background	Nonprofessional Background		
SCL	−1.26	−0.52	−0.77	−0.62	−0.48	−0.79	−0.53	−0.75	1.329	0.265
PA	1.42	0.5	0.71	−6	1.04	−0.91	−1.94	−1.78	3.584	0.014
NA	−3.54	−4.63	−2.73	−3.91	−0.85	−3.66	−0.54	−1.25	0.120	0.948

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
