# Peer review of "Effects of Plant Communities on Human Physiological Recovery and Emotional Reactions: A Comparative Onsite Survey and Photo Elicitation Study"

_ijerph, 2022, doi:10.3390/ijerph19020721_

Round 1

Reviewer 1 Report

The article is original in its approach and can provide results with practical consequences in the design of green spaces and gardens, but methodologically there is room for improvement.
Fundamentally:
- the age variable needs to be reported and included in the analyses. It is suggested to use analysis of covariance or multiple regression models.
- It would be desirable to incorporate at least a third modality in the data collection methods used (virtual reality) to improve the validity of the design. Although this is not possible once the research has been carried out, but at least indicate it as a suggestion for future research. 
- Improve the description of the sample, in the sense that in the categories used, we do not know whether students whose curriculum disciplines include mathematical calculation are included (remember that the stimulus used to increase stress is a numerical calculation test).
- Use another concept that is more appropriate than Mental Health. Perhaps it would be more consistent with the measurements made to refer to emotional reaction.

The comparison of the two perception methods, although seemingly well thought out, leaves some doubts about the methodological control of possible confounding variables to meet the requirements of an experimental design.

In addition, the method of showing photographs may be too "artificial" or "unrealistic" compared to perceiving the landscape in situ. Perhaps employing a third method, such as the use of virtual reality goggles, would give more information by providing a method of comparison that could perhaps be considered as an intermediate value between the two methods of perception employed and thus increase the validity of the experimental design.
There is one key variable that has not been included in the analysis and no information is given about it, the Age variable.
The age variable could be acting as an effect modifying variable and we cannot ascertain this.

Author Response

Thank you for your letter and for the reviewers’ comments concerning our manuscript entitled “Effects of plant communities on human physiological recovery and emotional reaction: A comparative onsite surveying and photo elicitation study” (ID: ijerph-1490846). Those comments are all valuable and very helpful for revising and improving our paper, as well as the important guiding significance to our researches. We have studied comments carefully and have made correction which we hope meet with approval. Revised portion are marked in red in the paper. The main corrections in the paper and the responds to the reviewer’s comments are as flowing:

Point 1: The age variable needs to be reported and included in the analyses. It is suggested to use analysis of covariance or multiple regression models. There is one key variable that has not been included in the analysis and no information is given about it, the Age variable.The age variable could be acting as an effect modifying variable and we cannot ascertain this.

Response 1: First, As Reviewer suggested that we lack a description of the age of the sample.Considering the Reviewer’s suggestion, We have made modifications and adjustments in three areas. therefore, we have supplemented the age information in the 235 lines of 2.2. Participants.

Line 235, the statements of “A total of 320 college students from various disciplines served voluntarily as participants in this experiment. Participants were divided into a professional group of students enrolled in landscape disciplines and a nonprofessional group of students enrolled in other disciplines.” were corrected as “A total of 320 college students (M age = 22.03, SD age = 2.13, age range = 18–25 years) studying various subjects were recruited as the participants in this study and were divided into a professional group that studied the subject of landscape architecture and a nonprofessional group that included students who studied other subjects (curriculum disciplines not including mathematical calculation).”

Second, the sample population selected in this study is undergraduate students. The age range of these samples is between 19-25 years old and the age span is small. Therefore, the significance of using covariance or multiple regression model analysis is not very obvious, but The age should indeed be described in the text, which is what we have missed.

Third, The age variable could be acting as an effect modifying variable. Therefore, in order to better verify the age variable, the sample will be further expanded in future studies, and discussions and research will be conducted to increase the richness of the experimental design. The content of this part is supplemented in the 589-592 lines of 5. Conclusions.

Line 589-592, “In addition, the age range of participants was between 19 and 25 years old, while other age ranges may also be different, and the age variable could act as an effect modifying variable. In future studies, the age variable must be further studied.” was added.

Point 2: It would be desirable to incorporate at least a third modality in the data collection methods used (virtual reality) to improve the validity of the design. Although this is not possible once the research has been carried out, but at least indicate it as a suggestion for future research. In addition, the method of showing photographs may be too "artificial" or "unrealistic" compared to perceiving the landscape in situ. Perhaps employing a third method, such as the use of virtual reality goggles, would give more information by providing a method of comparison that could perhaps be considered as an intermediate value between the two methods of perception employed and thus increase the validity of the experimental design.

Response 2: We have made correction according to the Reviewer’s comments.

First, thank the reviewers for their comments, and secondly, virtual reality technology could perhaps be considered as an intermediate value between the two methods of perception employed and thus increase the validity of the experimental design. This is a shortcoming of our research, so , We have supplemented the explanation in lines 598-599&602-605 of 5.Conclusions in our text, we will indicate it as a suggestion for future research.

Line 598-599,“in follow-up studies, we can consider the use of different forms of landscape perception and experience.”was added.

Line 602-605, “In addition, the method of showing photographs may be somewhat "single" compared to perceiving the landscape in situ. Perhaps employing a third method, such as the use of virtual reality goggles, could thus increase the richness of the experimental design.” was added.

Point 3: Improve the description of the sample, in the sense that in the categories used, we do not know whether students whose curriculum disciplines include mathematical calculation are included (remember that the stimulus used to increase stress is a numerical calculation test).

Response 3: First of all, We are very sorry for our negligence of did not describe the sample clearly and there was an error. Secondly, in the description of the professional background, it should be more clear that the type of the course subject, such as the student of mathematical calculation mentioned by the reviewer, and finally, We are very sorry for our incorrect writing numerical calculation test. So we are in the text Modifications and detailed descriptions of the professional background of the samples (2.2 Participants) were made to avoid loopholes.

Line 235-240, the statements of “A total of 320 college students from various disciplines served voluntarily as participants in this experiment. Participants were divided into a professional group of students enrolled in landscape disciplines and a nonprofessional group of students enrolled in other disciplines. The male-to-female ratio was 1:1, and the ratio of participants in the professional group to those in the nonprofessional group was 1:1.5. ” were corrected as “A total of 320 college students (M age = 22.03, SD age = 2.13, age range = 18–25 years) studying various subjects were recruited as the participants in this study and were divided into a professional group that studied the subject of landscape architecture and a nonprofessional group that included students who studied other subjects (curriculum disciplines not including mathematical calculation). The ratio of males to females was 1:1, and the ratio of professionals to nonprofessionals was 1:1.5.”

Line 250-251, the statements of “mathematical” were corrected as “numerical calculation”

Line 252, the statements of “mathematical operations” were corrected as “numerical calculation”.

Point 4: Use another concept that is more appropriate than Mental Health. Perhaps it would be more consistent with the measurements made to refer to emotional reaction.

Response 4: We have made correction according to the Reviewer’s comments.

Thanks to the reviewers for their comments. The concept of mental health is somewhat broad, and the changes in the mental state of participants cannot be reflected in the text. The mental state described in this text is mainly emotional changes, so mental health is changed to emotional response.

Line 28&142&304&353&467&472, the statements of “mental health” were corrected as “emotional reaction”.

Point 5: The comparison of the two perception methods, although seemingly well thought out, leaves some doubts about the methodological control of possible confounding variables to meet the requirements of an experimental design.

Response 5: We have made correction according to the Reviewer’s comments.

First, thanks to the reviewers for their comments. Secondly, compared to indoor experiments, outdoor experiments really cannot control all uncontrollable confounding variables like indoor experiments. However, outdoor experiments are tests that must be carried out. We try our best to minimize the uncontrollable factors of confounding variables. For example, outdoor temperature, humidity, radiation, irradiation angle, light, wind speed, and sound are all controlled to the utmost extent. Finally, we added the content on the basis of lines 206-208 of the original text 2.2.2. Procedure Part (In addition, before the start of each test, a reminder was posted within 2 meters of the test site to inform visitors of the test area, thereby reducing interference from external factors (such as visitors’ activities, noise, etc.)) of the content (lines. 210-212&pp.220-221).

Line 210-212, “To reduce the occurrence of confounding variables, we ensured that the surrounding environment was quiet while keeping the light, temperature, humidity, and wind speed in the landscape area similar. ” was added

Line 220-221, “Photo-elicitation was conducted in quiet and well-lit classrooms.” was added.

Reviewer 2 Report

Some of the comparable studies quoted might find a better place in the introduction. Certainly authors must have made some choices but the motivation can be a bit clearer. 

Author Response

Thank you for your letter and for the reviewers’ comments concerning our manuscript entitled “Effects of plant communities on human physiological recovery and emotional reaction: A comparative onsite surveying and photo elicitation study” (ID: ijerph-1490846). Those comments are all valuable and very helpful for revising and improving our paper, as well as the important guiding significance to our researches. We have studied comments carefully and have made correction which we hope meet with approval. Revised portion are marked in red in the paper. The main corrections in the paper and the responds to the reviewer’s comments are as flowing:

Point 1: Some of the comparable studies quoted might find a better place in the introduction. Certainly authors must have made some choices but the motivation can be a bit clearer. â€¨

Response 1: We have made correction according to the Reviewer’s comments.

First, thank the reviewers for their comments. Secondly, the text 1. Introduction is divided into two parts, the case study description about landscape perception and the case description of landscape experience, 1.1. Two visual experiences: onsite survey and image perception, which mainly describes the vision Experience-related case studies, 1.2. Different types of landscape perception experience, mainly describe related case studies of different landscape types (forests, parks, woodlands, etc.) experiences, as well as case studies of the use of two physical and psychological indicators of SCL and PANAS. Finally, based on the reviewer’s comments and our organization and adjustment of the introduction, in lines 37 to 43, in order to support the importance of environmental health, a paragraph reviewing human well-being and Research on the relationship between green urban areas. Also, in order to make the front and back logic flow smoothly, move the Sevenant and Antrop study in lines 80-82 to lines 75-78. And, in order to enrich the case studies of visual experience, over the last few years there is the Virtual Reality assessment tool, we added a part of "the study of the relationship between human well-being" in lines 37 and 84-87. and green urban areas" and "VR technology research".

Line 38-43, “For people living in an urban environment, urban green space is an important element of well-being, but it is often in short supply. One important element for resident well-being and quality of life is the availability of urban green space. There are different ways in which urban green space can positively influence well-being and health[26]. Benefits can accrue from increased activity levels as a result of being in contact with nature[24]. ” was added.

Line 83-86, “There are also researchs showed significant differences in their preferences for the urban green spaces with the different vegetation structures through Virtual Reality technology, and the semi-open green space received the highest preference score[12].” was added.

Line 75-78, “Sevenant and Antrop found the effectiveness of using photographs instead of real landscapes for landscape assessment to be influenced by issues such as perspective and the quality of the photographs taken [3].” was added.

Line 84-86, “Other studies have shown significant differences in their preferences for urban green spaces with different vegetation structures through virtual reality technology, and semiopen green spaces receive the highest preference score [12].” was added.

Reviewer 3 Report

The article describes. a study of landscape perception on selected examples through participant observation and photo observations. The content falls under the profile of Int. J. Environ. Res. Public Health, but needs improvement.

In the chapter presenting the methodology, some elements should be clarified. Test criteria are not entirely clear. Additional description would be needed to determine the vegetation period in which the study was conducted. Selected landscape study areas look different during the vegetation period and outside the vegetation period (winter). Specific dates would need to be clarified and a more accurate description of what climate the field surveys were conducted in and photos taken for the remote surveys. Also, section 2.3 on stress should be refined and explain in more detail why this factor was introduced into the study. I suggest looking at the following publications:

Zhang, L.; Liu, S.; Liu, S. Mechanisms Underlying the Effects of Landscape Features of Urban Community Parks on Health-Related Feelings of Users. Int. J. Environ. Res. Public Health 2021, 18, 7888. https://doi.org/10.3390/ijerph18157888

Lee, Min-Sun et al. “Interaction with indoor plants may reduce psychological and physiological stress by suppressing autonomic nervous system activity in young adults: a randomized crossover study.” Journal of physiological anthropology vol. 34,1 21. 28 Apr. 2015, doi:10.1186/s40101-015-0060-8

The results are described in a very broad and correct manner. The statistics are not objectionable. I miss the reference to the plant vegetation criterion I mentioned above. The discussion is done thoroughly and does not raise concerns.

The conclusions should be expanded and made more specific referring to the results of the study. Right now they are a little too general.

Author Response

Thank you for your letter and for the reviewers’ comments concerning our manuscript entitled “Effects of plant communities on human physiological recovery and emotional reaction: A comparative onsite surveying and photo elicitation study” (ID: ijerph-1490846). Those comments are all valuable and very helpful for revising and improving our paper, as well as the important guiding significance to our researches. We have studied comments carefully and have made correction which we hope meet with approval. Revised portion are marked in red in the paper. The main corrections in the paper and the responds to the reviewer’s comments are as flowing:

Point 1: In the chapter presenting the methodology, some elements should be clarified. Test criteria are not entirely clear. Additional description would be needed to determine the vegetation period in which the study was conducted. Selected landscape study areas look different during the vegetation period and outside the vegetation period (winter). Specific dates would need to be clarified and a more accurate description of what climate the field surveys were conducted in and photos taken for the remote surveys.ʉ۬

Response 1: We have re-written this part according to the Reviewer’s suggestion.

First, thank the reviewers for their comments. Secondly, we only mentioned the time of outdoor experiments in the text, but ignored the vegetation period. This is our mistake. Secondly, during the vegetation period and outside the vegetation period (winter), it is true that the morphological characteristics of plants and the effects of perception and experience on the human body are different. We took this into consideration, but we did not describe it clearly in the text. Finally, we wrote in text 2.2.3 has been added to Experimental design, which describes the different vegetation periods of outdoor experiments, as well as the photo shooting time and indoor photo stimulation experiment time, to fill in the lack of this part.

Line 171, “This study was conducted in October 2019 around the autumnal equinox of the traditional Chinese solar terms. ” was deleted.

Line 222-228, “2.2.3 Vegetation period

The selected landscape study area is within the vegetation period (autumn). The vegetation period in the onsite surveys was approximately October 10, 2019, and the test time was between October 10 and October 20. The photos taken for the remote surveys were from October 11 to October 18, 2019. The photo elicitation experiment was completed in quiet and well-lit classrooms from November 2, 2019, to November 15, 2019.” was added.

Point 2: Also, section 2.3 on stress should be refined and explain in more detail why this factor was introduced into the study.Also, section 2.3 on stress should be refined and explain in more detail why this factor was introduced into the study. I suggest looking at the following publications:

Zhang, L.; Liu, S.; Liu, S. Mechanisms Underlying the Effects of Landscape Features of Urban Community Parks on Health-Related Feelings of Users. Int. J. Environ. Res. Public Health 2021, 18, 7888. https://doi.org/10.3390/ijerph18157888

Lee, Min-Sun et al. “Interaction with indoor plants may reduce psychological and physiological stress by suppressing autonomic nervous system activity in young adults: a randomized crossover study.” Journal of physiological anthropology vol. 34,1 21. 28 Apr. 2015, doi:10.1186/s40101-015-0060-8

Response 2: We have re-written this part according to the Reviewer’s suggestion.

First, thank the reviewers for their comments. Secondly, we also read and browsed these two texts based on the reviewers’ recommendations. The reason why we introduced the stress factors in Section 2.3 is that in order to more clearly reflect the participants to experience the physiological recovery and emotional conditions before and after the experience, we use the noisy environment to simulate the outdoor urban environment, which is used to increase the stress level of the human body. When the stress level rises, when the participants are experimenting with ornamental plants, the stress level changes The trend (reduce pressure or increase pressure) will be clearer and more obvious. Finally, we added the description of this part in the text 2.4. Stress task lines 251-255.

Line 251-255, “We told the participants that the purpose of this experiment was to evaluate their performance in numerical calculations, and we scored and ranked their performance to reflect the participants’ physiological recovery and emotional changes more clearly after the experience. We used numerical calculations and simulations of noisy environments to stimulate psychological and physiological responses.” was added.

Point 3: The results are described in a very broad and correct manner. The statistics are not objectionable. I miss the reference to the plant vegetation criterion I mentioned above. The discussion is done thoroughly and does not raise concerns.The conclusions should be expanded and made more specific referring to the results of the study. Right now they are a little too general.

Response 3: We have re-written this part according to the Reviewer’s suggestion.

First of all, thank the reviewers for their comments. Secondly, the research conclusions and research deficiencies include specific and extended descriptions. At the same time, the research conclusions are expanded based on the research results and the research conclusions are made concrete. Finally, we will The original content of 4.5 Limitations has been adjusted to 5.Conclusions. At the same time, we added a part of the content in lines 577-579&588-606 of 5.Conclusions and revised the content of lines 584-587.

Line 577-579, “Participants watched the lawn community in the form of onsite survey, which reduced SCL, and at simultaneously had a better inhibitory effect on negative emotions.” was added.

Line 584-587, the statements of “It is recommended that the combination of plant community types should be reasonably used in future landscape design according to the onsite environment.” were corrected as “Thus, the approach to landscape perception should be carefully selected for a specific landscape during a specific season to provide a scientific basis for evaluating landscape perception and preference in the future.”

Line 588-606, “However, this study has some weaknesses. First, all the participants were college students. Many other investigations have also used college students as the target population. However, the results may not fully reflect other social groups. In addition, the age range of participants was between 19 and 25 years old, while other age ranges may also be different, and the age variable could act as an effect modifying variable. In future studies, the age variable must be further studied. Therefore, the sample population should be expanded in subsequent studies. Second, our study used the model of stopping to admire a plant landscape, in which the behavioral patterns of participants’ appreciation of the scenery are mostly stopping and touring, which is more consistent with the ways in which visitors experience landscapes when recreating in parks. Therefore, in follow-up studies, we can consider the use of different forms of landscape perception and experience. Finally, this study was only conducted in the autumn, but the preferences of the participants may be affected by the changes in weather and seasonal changes in plants, especially through the perception method of onsite surveying. The landscapes of plant communities during different seasons can be further studied and explored. In addition, the method of showing photographs may be somewhat "single" compared to perceiving the landscape in situ. Perhaps employing a third method, such as the use of virtual reality goggles, could thus increase the richness of the experimental design.” was added.

Reviewer 4 Report

This is an interesting study investigating two different methodological approaches in the effects of plant communities on human physiological recovery 2 and mental health. However, the title of the text is not suitable. In my opinion, you should mention the methods that you examine, since the research is not actually about the effects of plant communities on human physiological recovery and mental health, but about the comparison of the two methods and their effectiveness. Also, please find attached further comments and suggestions to improve your manuscript in the attached file.

Author Response

Thank you for your letter and for the reviewers’ comments concerning our manuscript entitled “Effects of plant communities on human physiological recovery and emotional reaction: A comparative onsite surveying and photo elicitation study” (ID: ijerph-1490846). Those comments are all valuable and very helpful for revising and improving our paper, as well as the important guiding significance to our researches. We have studied comments carefully and have made correction which we hope meet with approval. Revised portion are marked in red in the paper. The main corrections in the paper and the responds to the reviewer’s comments are as flowing:

Point 1: The title of the text is not suitable. â€¨

Response 1: It is really true as Reviewer suggested that since the research is not actually about the effects of plant communities on human physiological recovery and mental health, but about the comparison of the two methods and their effectiveness. So we modified and adjusted the title of the text.

Line 2-4, the statements of “Effects of plant communities on human physiological recovery and mental health: A comparative field study”were corrected as “Effects of plant communities on human physiological recovery and emotional reactions: A comparative onsite survey and photo elicitation study”.

Point 2: Please consider including a paragraph in your introduction reviewing studies on the relationship between human well-being and green urban areas, such as: Votsi, N. E. P., Mazaris, A. D., Kallimanis, A. S., Drakou, E. G., & Pantis, J. D. (2014). Landscape structure and diseases profile: associating land use type composition with disease distribution. International journal of environmental health research, 24(2), 176-187, or Bertram, C., & Rehdanz, K. (2015). The role of urban green space for human well-being. Ecological Economics, 120, 139-152, and other, in order to support the importance of environmental health, before investigating the landscape perception and the experience research.

Response 2: We have made correction according to the Reviewer’s comments.

First, thank the reviewers for their comments. Secondly, we read and browsed the two texts suggested by the reviewers. In the 1. Introduction section, our text describes the case study from the beginning of the investigation of landscape perception and experience research, which is missing for the study of the relationship between human well-being and green urban areas, finally, we added reviewing studies on the relationship between human well-being and green urban areas in lines 38-43 of 1.Introduction.

Line 38-43, “For people living in an urban environment, urban green space is an important element of well-being, but it is often in short supply. One important element for resident well-being and quality of life is the availability of urban green space. There are different ways in which urban green space can positively influence well-being and health[26]. Benefits can accrue from increased activity levels as a result of being in contact with nature[24].” was added.

Point 3: Except for these two methods, over the last few years there is the Virtual Reality assessment tool, which should also be mentioned here, since you're stating the visual experiences.

Response 3: We have made correction according to the Reviewer’s comments.

First of all, thank the reviewers for their comments. Second, when writing the text, we have considered adding a case study of VR technology, but we are worried about whether it is feasible. Finally, we based on the reviewers’ comments and our own understanding. 1.1Two visual experiences: onsite survey and image perception Lines 84-86 add a case study of visual experience of VR technology.

Line 84-86, “Other studies have shown significant differences in their preferences for urban green spaces with different vegetation structures through virtual reality technology, and semiopen green spaces receive the highest preference score [12].” was added.

Point 4: Please define what you mean here, in your study, by the term "plant communities". Parks do not include plant communities? Actually, all green urban areas are composed of plant communities...what is the difference in your case study?

Response 4: First, thank the reviewers for their comments. Secondly, the “plant community” in this study refers to the type of space creation that combines the three levels of vegetation structures of trees, shrubs, and herbs, such as a single-layer that creates an open space. Grassland, single-layer woodland and tree-grass composite woodland for semi-open space (underforest, overstory landscape), and tree-shrub-grass composite woodland for closed space. Finally, the reason why the case studies in the text want to express is that among the many research objects, the main researches are forest land, parks, gardens, wetlands and other landscape types. However, the landforms of each type of green space are very different. There are many types of space creation with different combinations of vegetation structures, and this study selects the more common ones with higher preference.

Point 5: Sections 4.1, 4.2, 4.3 should be limited or even merged. There is no need for so extensive writing, it is difficult to read and stay focused, whereas all these effects could be mentioned in a more condensed way.

Response 5: We have made correction according to the Reviewer’s comments.

First, thank the reviewers for their comments. Secondly, when we wrote the text, we considered even merged the three aspects of 4. Discussion, but found that the content after even merged was too confusing. Therefore, in order to compile the research results The analysis of the three aspects of the problem is clear. The content of 4. Discussion in the text is divided into three aspects to describe. Finally, we delete and organize the content of the discussion part based on the comments of the reviewers.

Line 434, “Although the landscape in a photo can be very attractive, the perception of the landscape and its surrounding environment cannot be reflected in this form.” was deleted.

Line 438, “This finding indicates that even if a complete image is presented through continuous photographs, some onsite information cannot be presented through images [20]. In the perception process during photo elicitation, the attributes of the visual perception experience of the participants are not diverse, and the participants cannot feel the sense of spatial integration of the onsite environment and can only outline a more subjective perceptual picture through self-awareness, which affects the perceived experience of the participants.” was deleted.

Line 440, the statements of “photo viewing” were corrected as “Site investigation”.

Line 440, “reduces the interaction and perceptual experience between the participants and the environment. These plant community types are all familiar in the daily lives of the participants. However, the participants cannot obtain the environmental information of the real scenes through the photos and thus may experience them based on their subjective cognition.” was deleted.

Line 460, “Therefore, the body in the landscape will produce a variety of sensory experiences, which can help the participants perceive and experience the landscape from multiple aspects, but the photo elicitation method lacks this advantage and can only present one sensory experience at a time. ” was deleted.

Line 475, “which to a certain extent supports the “savanna hypothesis” [57]. ” was deleted.

Line 480, “In traditional Chinese culture, coniferous plants hold a solemn and sacred cultural connotation [60]. Wang (2017) also showed that the height of woodland is related to preference. Planting medium and low vegetation along the road, avoiding tall vegetation, will attract more visitors to watch and enter a woodland [59].” was deleted.

Line 491, “However, excessively dense trees will limit the field of view and reduce the sense of security [64]. Therefore, the combination of plant community types in landscape design should be carefully selected.” was deleted.

Line 493, “Changes in the trends of physiological recovery (SCL) and mental health recovery (PA and NA) can be divided into two stages and three changes. The two stages are the stress stage and the immersion stage, and the three changes are the stress value (â–³T1), immersion value (â–³T2), and recovery value (â–³T3). In the stress stage of our experiment (corresponding to the change in stress value, â–³T1), the stress values measured by SCL (â–³T1) were significantly higher than the baseline (T0), and the NA scores were also significantly higher than the PA scores. ” was deleted.

Line 503, “In the immersion stage (corresponding to changes in the immersion value â–³T2),” was deleted.

Line 522, “Another study used photo elicitation to examine the difference in perceived safety in terms of gender. For men and women, vegetation was considered to be moderately safe, and men’s perception of safety was higher than that of women [43].” was deleted.

Line 523, “most of the cognitive differences among disciplines are attributed to lack of information. ” was deleted.

Line 525, “Nonprofessional, less educated visitors, visitors from rural areas, and visitors to the landscape for the first time often scored higher.” was deleted.

Line 432, the statements of “Advantages and significance” were corrected as “Future directions”

Round 2

Reviewer 1 Report

The authors have adequately addressed the objections raised in the first review.

The study, due to the type of design used and the methods compared, is difficult to generalise or extrapolate to the rest of the general population.
Within its limitations, it does provide some descriptive value on how people perceive the landscape.